# Data Analysis and Systematic Scoping Review on the Pathogenesis and Modalities of Treatment of Thyroid Storm Complicated with Myocardial Involvement and Shock

**DOI:** 10.3390/diagnostics13193028

**Published:** 2023-09-23

**Authors:** Eman Elmenyar, Sarah Aoun, Zain Al Saadi, Ahmed Barkumi, Basar Cander, Hassan Al-Thani, Ayman El-Menyar

**Affiliations:** 1Faculty of Medicine, Internship, Bahcesehir University, Istanbul 34734, Turkey; eman.elmenyar@bahcesehir.edu.tr (E.E.); sarah.aoun@bahcesehir.edu.tr (S.A.); khudhur.alsaadi@bahcesehir.edu.tr (Z.A.S.); ahmed.barkumi1@bahcesehir.edu.tr (A.B.); 2Department of Emergency Medicine, Kanuni Sultan Süleyman Training & Research Hospital, Istanbul 34303, Turkey; basarcander@yahoo.com; 3Department of Surgery, Trauma Surgery, Hamad General Hospital, Doha 3050, Qatar; althanih@hotmail.com; 4Department of Surgery, Trauma Surgery, Clinical Research, Hamad General Hospital, Doha 3050, Qatar; 5Department of Clinical Medicine, Weill Cornell Medical College, Doha 24144, Qatar

**Keywords:** thyroid storm, thyrotoxic crisis, hyperthyroidism, myocardial disease, shock, mechanical support, thyroidectomy, beta blockers, Burch–Wartofsky score, tachyarrhythmia, heart failure, gender, mortality

## Abstract

Thyroid storm (TS) is a rare and fatal endocrine emergency that occurs due to undiagnosed and inadequately treated hyperthyroidism after stressful conditions in patients with thyroid disorders. The objective of this systematic scoping review was to better understand the pathophysiology of TS and its complications, in terms of myocardial affection, tachyarrhythmia, and cardiogenic shock. In addition, we explored the pharmacological, mechanical, and surgical treatments for TS. We also evaluated the outcomes of TS according to sex and cardiac involvement. Additionally, analytical analysis was performed on the selected data. A literature review of peer-reviewed journals was carried out thoroughly using medical terms, MeSH on PubMed, Google Scholar, and combinations such as thyrotoxicosis-induced cardiomyopathy, thyroid storm, cardiogenic shock, myocardial infarction, endocrine emergency, Burch–Wartofsky score, extracorporeal circulatory support, and thyroidectomy. A total of 231 papers were eligible (2 retrospective studies, 5 case series, and 224 case reports) with a total of 256 TS patients with cardiac involvement between April 2003 and August 2023. All age groups, sexes, patients with TS-induced cardiomyopathy, non-atherosclerotic myocardial infarction, tachyarrhythmia, heart failure, shock, and different forms of treatment were discussed. Non-English language articles, cases without cardiac involvement, and cases in which treatment modalities were not specified were excluded. Female sex was predominant, with 154 female and 102 male patients. Approximately 82% of patients received beta-blockers (BBs), 16.3% were placed on extracorporeal membrane oxygenation (ECMO) support, 16.3% received therapeutic plasma exchange (TPE), and 13.8% underwent continuous renal replacement therapy (CRRT), continuous venovenous hemofiltration (CVVHD), or dialysis. Overall, 18 females and 16 males died. BB-induced circulatory collapse, acute renal failure, CRRT, and ventricular fibrillation were significantly associated with mortality. Awareness of TS and not only thyrotoxicosis is vital for timely and appropriate treatment. The early diagnosis and management of TS in cardiac settings, including pharmacological, mechanical, and surgical modalities, can save high-risk patients. Sex matters in the presentation, treatment, and mortality of this population. However, further large-scale, and well-designed studies are required.

## 1. Introduction

Thyroid storm (TS) is a rare, serious endocrine emergency that occurs as a consequence of undiagnosed or inadequately treated hyperthyroidism [1]. It is an extreme form or crisis form of thyrotoxicosis that complicates thyroid disease in approximately 1–5.4% of hospitalized patients [2,3]. However, in a few reports, it may reach 10% [4]. In Japan, the annual incidence of TS in hospitalized patients is 0.20 per 100,000 individuals [3]. In Taiwan, the incidence is 0.55 per 100,000 persons per year and 6.28 per 100,000 hospitalized patients per year, with a 90-day mortality rate of 8.12% [5]. A study from the USA showed an incidence of 0.57–0.76 cases/100,000 persons per year, and 4.8 and 5.6/100,000 hospitalized patients per year [6]. It mostly affects females and patients with underlying Graves’ or autonomous nodular disease [7]. The diagnosis of TS is mainly based on clinical findings and an early high index of suspicion, as there are no specific laboratory findings, no thyroid function test cutoffs, or widely accepted diagnostic criteria [2]. TS usually presents with multi-organ dysfunction depending on the rapidity of the available free nonbound fraction of the thyroid hormone in circulation and its exaggerated effect on body tissues and organs [2]. The adrenergic crisis with abrupt responsiveness to the endogenous catecholamines during the thyrotoxic status has worse outcomes in patients with TS [8]. A wide variety of clinical manifestations are caused by the hypermetabolic effects of thyroxine hormones in different organs of the body. These manifestations include generalized symptoms such as fever, fatigue, and weight loss, as well as thermoregulatory dysfunction such as sweating and heat intolerance. Central nervous system dysfunction is characterized by numbness, tremors, and anxiety. Cardiovascular findings include hypertension, tachyarrhythmia (sinus tachycardia, atrial fibrillation, atrial flutter, supraventricular tachycardia, and ventricular arrhythmia), myocarditis, myocardial infarction, heart failure, cardiogenic shock (CS), and cardiac arrest. Gastrointestinal-hepatic system symptoms include jaundice, abdominal pain, diarrhea, vomiting, and nausea. Other systematic findings such as dyspnea, exophthalmos, anterior neck swelling, and leg edema have been reported. Thyrotoxicosis-induced cardiomyopathy is observed in <1% of patients, where overt CS develops because of severe ventricular systolic function impairment [9]. However, ventricular changes are reversible if treatment to correct thyroid hormone dysfunction is administered early and appropriately [10].

The presence of thyrotoxicosis is made according to the elevation of free Triiodothyronine (free T3) to >8.3 mU/L and free thyroxine (free T4) to >19.4 mU/L and a low or undetectable thyroid-stimulating hormone (TSH) level of < 0.05 mU/L [11]. Notably, the standard thyroid function tests cannot differentiate between hyperthyroidism and TS. Moreover, the degree of hyperthyroidism is not a criterion for TS diagnosis [12].

Diagnosis of TS and its criteria: The diagnosis of TS is based on various factors [13]. In most cases, the occurrence of TS requires superimposed factors such as thyroid surgery, interrupted thyroid treatment, in-hospital infection, stressful conditions, and major surgery or trauma. However, a clear precipitating factor can be missed in up to 40% of the TS cases [2,14]. TS is a thyrotoxic status with at least one CNS manifestation plus one or more other symptoms, such as thermoregulatory dysfunction, cardiovascular findings in terms of tachycardia, atrial fibrillation, congestive heart failure, gastrointestinal and hepatic dysfunction, and precipitant history [15]. ‘OR’ A combination of at least three features among fever, GI/Hepatic, CHF, or tachycardia [16]. The Burch–Wartofsky point scale (BWS) is a quantitative diagnostic scoring system used to identify the presence of TS if the patient’s score is > 45 points [15]. However, the scoring systems are mainly for guidelines. In this review and data analysis, we relied only on the BWS criteria for selected cases. The diagnosis of TS is confirmed when there are various systemic clinical findings [17]. BWS is a quantitative diagnostic scoring system used to identify TS. It accounts for an array of clinical manifestations in the major body systems related to TS diagnosis of TS. A BWS score of 45 or greater is indicative of an ST, a score between 25 and 44 is suggestive of impending TS, and a score >25 is suggestive of (pending storm).

The most striking symptom of TS is fever >102 °F, and when combined with other thyrotoxic symptoms, immediate care should be provided [18]. Moreover, fever is often associated with the mortality rate is high, ranging from 8% to 25% or even 30%, despite early treatment; therefore, urgent aggressive treatment is mandatory [4,19]. The most common causes of death in patients with TS are multi-organ failure, heart failure, dysrhythmia, respiratory failure, and sepsis. Most information on TS stems from case reports or series; therefore, well-established guidelines are lacking. However, in patients with severe hyperthyroidism, physicians should search for TS criteria immediately after implementing lifesaving measures. In this review, we aimed to explore myocardial involvement in terms of injury, dysrhythmia, cardiomyopathy, failure, and CS during TS and the modalities of treatment and their efficiency, including pharmacological, mechanical, and surgical options. We also addressed the objectives based on the pathogenesis, patient sex, and outcomes.

## 2. Materials and Methods

*Objectives:* The first objective of this review aimed to explore the occurrence, pathophysiology, and pattern of myocardial affection due to TS in males and females. Second, we evaluated the pharmacological, mechanical, and surgical treatments of TS in these populations. Finally, the clinical outcomes of TS, cardiac complications, and patient management were evaluated.

*Protocol*: this review followed the preferred reporting items for systematic reviews and meta-analyses extension for scoping reviews (PRISMA-ScR) checklist (http://www.prisma-statement.org/documents/PRISMA-ScR-Fillable-Checklist) (accessed on 31 July 2023) (Appendix A). Figure 1 shows the scoping review design according to the PRISMA items.

*Data collection*: A systematic scoping review of peer-reviewed articles published between April 2003 and August 2023 was conducted. Additionally, analytical analysis was performed on the selected data. A thorough search was performed using medical terms and combinations of thyrotoxicosis crisis, thyrotoxicosis-induced cardiomyopathy, thyrotoxic cardiomyopathy, Takotsubo cardiomyopathy (TCM), thyroid storm, endocrine emergency, heart failure, BWS, CS, circulatory collapse, tachyarrhythmias, non-atherosclerotic myocardial infarction, heart failure, organ failure, ejection fraction (EF%), extracorporeal membrane oxygenation (ECMO), extracorporeal mechanical support, intra-aortic balloon pump (IABP), Impella device, left ventricular assist device (LVAD), therapeutic plasma exchange (TPE), continuous renal replacement therapy (CRRT), thyroidectomy, and radioiodine ablation (RAI) in electronic databases, such as PubMed, Google Scholar, and MeSH on PubMed search engines.

*Eligibility criteria*: To allocate eligible studies, a comprehensive search strategy using keywords and text words highlighting variable treatments and interventions was used. Data included prospective and retrospective studies, case reports, and case series. Abstract: Fully pertinent details have been included such as age groups, sex, cardiac involvement, and different forms of pharmacological, mechanical, and surgical treatment. Exclusion criteria were articles in languages other than English unless the abstract entailed all the required information, cases of TS without cardiac involvement, and cases where treatment modalities were not specified. Abstracts, systematic reviews, and meta-analyses in which relevant individual cases were not identified or missing data were excluded. Moreover, any cases of thyroid disorders without a clear statement of “thyroid storm” or BWS (or criteria for score calculation) were excluded (see the design of the review in Figure 1).

*Information sources and search methods*: A comprehensive search of electronic databases such as PubMed/Medline and Google Scholar was performed between April 2003 and August 2023. Articles with information on the treatment modalities used for cases of cardiac manifestations due to TS, such as cardiomyopathy, infarction, tachyarrhythmias, heart failure, and CS, were included in this review. We also emailed some authors for further clarification.

*Study Selection*: A systematic search was carried out using keywords.

***Data collection process***: A thorough manual search was performed of eligible articles, and duplicates were removed. EE, SA, ZA, and AB independently searched, screened, and extracted the articles, and a consensus for eligibility was reached by all and reviewed by AE. The information obtained was categorized within tables that included the author(s), number of cases per article, sex, mean age, history of thyroid disease, presentation and BWS, electrocardiogram (ECG), initial and follow-up ejection fraction (EF%), CS and cardiac arrest timing, type of beta-blocker (BB) used, use of cardiac devices, extracorporeal circulatory support or surgery including intra-aortic balloon pump (IABP), Impella device, left ventricular assist device (LVAD), therapeutic plasma exchange (TPE), extracorporeal membrane oxygenation and duration (ECMO) and CRRT, thyroidectomy, radioactive iodine ablation, presence of organ failure, and patient outcomes.

*Risk of bias assessment in individual and across studies*: The risk of bias assessment was not applicable, as all cases were collected from case reports, case series, and retrospective studies published in peer-reviewed journals; moreover, there were no other types of studies on this topic. However, this value was generally low. Bias across studies is at the level of the provider who executes management. This is based on their knowledge of the subject matter and preferred methods. No structured, large-scale studies have addressed the topic of this review.

*Definitions*: The presence of thyrotoxicosis and TS will be discussed in detail later. Cardiac manifestations were identified based on the patient’s clinical findings, laboratory values of troponin and B-type natriuretic peptides, ECG, chest radiographs, echocardiogram, electrocardiogram, coronary angiogram (if available), and serial transthoracic echocardiogram (echo) for heart failure findings. Myocardial affection included myocardial injury, ST- or non-ST-segment elevation myocardial infarction, acute heart failure, tachyarrhythmia, cardiomyopathy, and particularly dilated cardiomyopathy (DCM). Echocardiographic findings of cardiomyopathy include atrial and ventricular dilation, global hypokinesis, and biventricular dysfunction with low ejection fraction (EF). Shock is defined as low blood pressure and insufficient end-organ hypoperfusion. Cardiogenic shock (CS) has a low cardiac index, elevated ventricular filling pressure, and decreased mixed venous oxygen saturation. However, most published TS cases do not define CS and sometimes mention circulatory collapse or cardiopulmonary arrest.

*Synthesis of results*: Data are presented as proportions, medians, and interquartile ranges (IQR). Comparisons between the two groups (males and females, and alive and dead) were made using Student’s t-test for continuous variables and the chi-square test (or Fisher’s exact test) for categorical variables. Mann–Whitney U and Kruskal–Wallis tests were used to compare variables with non-normal distributions (non-parametric tests). A two-tailed *p* value ≤ 0.05 was considered statistically significant. Statistical Package for the Social Sciences (SPSS) for Windows (version 21.0; SPSS, Chicago, IL, USA) was used for data analysis.

*Summary Measures*: Principal summary was based on sex, management modalities, and outcome (death).

## 3. Results

Overall findings: A total of 256 patients who presented with TS-induced cardiac manifestations were retrieved from 231 studies. Of these, 5 articles were case series, 224 were case reports, and 2 were retrospective studies. Several articles did not specify a BWS score; however, the necessary criteria were mentioned throughout the articles. Hence, we manually calculated the BWS scores according to the presented criteria. Notably, 41 articles did not report BWS and had insufficient criteria for manual calculations. However, these articles were included because their authors defined cases as ‘thyroid storm’ or ‘thyrotoxic crisis. Table 1 and Table 2 show comparisons based on patient sex (154 females and 102 males) [1,9,10,12,19,20,21,22,23,24,25,26,27,28,29,30,31,32,33,34,35,36,37,38,39,40,41,42,43,44,45,46,47,48,49,50,51,52,53,54,55,56,57,58,59,60,61,62,63,64,65,66,67,68,69,70,71,72,73,74,75,76,77,78,79,80,81,82,83,84,85,86,87,88,89,90,91,92,93,94,95,96,97,98,99,100,101,102,103,104,105,106,107,108,109,110,111,112,113,114,115,116,117,118,119,120,121,122,123,124,125,126,127,128,129,130,131,132,133,134,135,136,137,138,139,140,141,142,143,144,145,146,147,148,149,150,151,152,153,154,155,156,157,158,159,160,161,162,163,164,165,166,167,168,169,170,171,172,173,174,175,176,177,178,179,180,181,182,183,184,185,186,187,188,189,190,191,192,193,194,195,196,197,198,199,200,201,202,203,204,205,206,207,208,209,210,211,212,213,214,215,216,217,218,219,220,221,222,223,224,225,226,227,228,229,230,231,232,233,234,235,236,237,238,239,240]. The median age of the patients included in this study was 42.5 years old with an interquartile range (IQR) of 31–58 and there was one patient aged 3 years. Female sex predominated (60%), with a median age of 43 yeas, whereas males (40%) had a median age of 42.

Three-quarters of the patients reported thyroid disorders. The most common diseases were Graves’ disease (28%), 11.4% hyperthyroidism/thyrotoxicosis (11.4%), drug-induced (11%), and nonspecific autoimmune thyroid disease (1%). Chronic thyroiditis or viral infection were observed in 4% of patients, whereas 7% of patients had no history of thyroid disease, and 26% had a specific thyroid disease.

The most common symptoms were dyspnea and palpitations. Other symptoms included nausea, vomiting, diarrhea, weight loss, sweating, cough, and chest pain. Based on echocardiogram results, the median initial EF was 25, with an IQR of (19–40). The latest EF reading during the follow-up was 50%, with an IQR of (36–58). However, the follow-up EF was not reported for 68 patients; 16 did not report on this because the patients died early, and the authors simply defined the patients’ EF as normalized or recovered.

The most common arrhythmias detected on electrocardiography (ECG) were atrial fibrillation (AF) (49%) and sinus tachycardia (28%), which are commonly detected in patients with cardiomyopathy. Approximately 15 patients showed no ECG findings. Other ECG findings included atrial flutter, ventricular fibrillation, ventricular tachycardia, multifocal atrial tachycardia (MAT), supraventricular tachycardia, pulseless electrical arrest, and asystole. A total of 14 patients presented with cardiac arrest on admission.

Of the total number of cases, 48% presented with multiple organ failure, of which the most common were heart failure (any degree of severity) 70%, hepatic failure (41.5%), and renal failure (20.7%). It is worth noting that most patients survived following admission and treatment; however, their death was either due to complications or treatment failure.

Gender and storm: There are no studies that have examined the impact of sex on the incidence of TS, its complications, treatment, and outcomes. Akamisu et al. [3] reported that the male-to-female ratio was 30/76, 74/204, and 15/59 in the overall literature and in Japanese TS1 and TS2 populations, respectively. In the USA, Waqar et al. observed that the male: female ratio was 25/75, 37/63, 29/71 and 25/75 for overall TS, associated ischemic events, acute HF, and arrhythmias, respectively [241]. In our analysis, 60% of the patients were female, whereas mortality was slightly higher in males. Table 1 and Figure 2 show comparisons between males and females with sustained TS and cardiac involvement, respectively. There were no significant statistical differences between males and females in terms of the cause of TS, such as Graves’ disease (30% vs. 26%), thyroiditis (7% vs. 2%), drug-induced (3% vs. 1.3%), and autoimmune (1% vs. 0.6%). Females had significantly higher rates of high BNP values, sinus tachycardia, high initial LVEF, pulmonary edema on admission, and Takatsubo cardiomyopathy. Males had significantly higher rates of AF on admission, liver failure, CHF (except for pulmonary edema), and dilated cardiomyopathy.

Management and Outcome: The treatment of TS involves targeting several mechanisms in terms of inhibiting new thyroid hormone synthesis, inhibiting thyroid hormone release, inhibiting the peripheral effect of thyroid hormone (20% of T3 is secreted from the gland and 80% is produced by conversion of T4 to T3), inhibiting enterohepatic circulation of thyroid hormones, inhibiting deiodinases D1 and D2, blunting the adrenergic surge, and supportive, resuscitative, and circulatory support measures, plasma exchange, and surgical treatment [2]. Management involves various modalities, including pharmacological, mechanical, and surgical options.

It was concluded that 91% of patients with cardiac involvement secondary to TS were administered anti-thyroid drugs (ATD). Additionally, 43% were administered inorganic iodide, such as potassium iodide one hour after ATD administration. As an additional treatment, 13% were administered cholestyramine and 70% were administered corticosteroids such as hydrocortisone or dexamethasone.

Regarding cardiac medications, 82% were given BBs as non-selective (i.e., propranolol and sotalol) or cardioselective (i.e., metoprolol and bisoprolol) BBs, in addition to short-acting cardioselective with short half-life (i.e., landiolol and esmolol). Calcium channel blockers were reported in 12.6% of cases. Moreover, 10.4% of patients were administered digoxin, and 33.6% of patients received inotropic medications and vasopressors. In addition, the patients received amiodarone.

Mechanical treatment, mainly supportive of pharmacological therapy or bridge to definitive surgery; 16.3% were placed on ECMO. The other commonly used modality was TPE in 16.3% of the patients. Moreover, 13.8% of patients underwent CRRT. Five patients had an Impella device installed, and 9.3% had an intra-aortic balloon pump. Chaaya et al., Ahmad et al., and Korte et al. reported the use of a left ventricular assist device [43,73,237]. In terms of other thyroid-related procedures, 18.6% of the patients underwent thyroidectomy, whereas 3.3% received RAI.

Table 2 and Figure 3 show the differences in treatment modalities between males and females. Female patients received more BBs, amiodarone, CRRT, and RAI, whereas male patients received more inotropes, vasopressors, steroids, ICD, ECMO, IABP, TPE, and thyroidectomies.

Figure 3 shows the types of beta blockers used in males and females after TS. Propranolol was the most prescribed agent in both sexes, with a slight increase observed in females.

The overall mortality rate was 13.5% (18 females and 16 males). Table 3 shows comparisons between alive and dead patients after cardiac TS. Deceased patients were significantly more likely to be male, develop ventricular fibrillation (VF) and circulatory collapse after receiving BBs, develop acute renal failure, and undergo CRRT or ECMO than live subjects, whereas thyroidectomy was associated with significantly higher rates of survival.

## 4. Discussion

This is a unique systematic scoping review exploring the pathophysiology and management of TSs complicated by cardiac involvement between 2003 and 2023. In addition, data analysis was performed for 256 patients with TS and cardiac involvement. TS is a thyrotoxicosis caused by an uncontrollable rise in endogenous thyroid hormones, leading to an exaggeration of wide clinical systematic presentations. The cardiovascular system is one of the organs that is most affected by TS. It has been noted that the cardiovascular events (4% acute ischemia, 14% acute HF) that developed in TS were significantly associated with increased patient age [241]; however, the comorbidities were also high in these patients. In an 18-year study period using the Japanese criteria for TS, Bourcier et al. [242] reported 92 TS cases with cardiac complications, such as CHF (72%), cardiac arrest before admission to the ICU (15%), SVT (60%), and VF (13%). A study from the USA reported supraventricular arrhythmias, ventricular arrhythmias, cardiac arrest, CS, CHF, and acute coronary syndrome (ACS) as 27.4% vs. 20%, 2.5% vs. 1.2%, 1.3% vs. 0.1%, 1% vs. 0.3%, 19.4% vs. 10.3%, and 1.8% vs. 0.7% in TS versus thyrotoxicosis without a storm, respectively [6]. In our analysis, we observed CHF (47%), CS (32%), VF (5%), AF (49%), and AMI (10%).

Precipitating factors: TS can be triggered by various factors; some may spark underlying thyroid dysfunction, and others exacerbate existing thyroid diseases, particularly Graves’ disease [16]. These precipitating factors include noncompliance with ATD, surgery (thyroid and non-thyroid), trauma, acute illness with infection, childbirth, hydatidiform mole, acute stress, drugs, and excess iodine intake [2,16,99,107,136,144]. As 99% of thyroid hormones are in a bound form, stressors may decrease the hormone-binding capacity and thus abruptly increase the concentration of the free hormones, leading to TS (2191). Even microtrauma in stable thyroid disorders, such as fine-needle aspiration (FNA), can cause TS [243].

Chen et al. [155] presented the case of a 46-year-old female who fell on the stairs, causing abrasions on the anterior aspect of her neck. Upon examination, the patient was diagnosed with TS complicated by intractable heart failure. Conte et al. [122] presented a case of strangulation that induced TS in a patient with no history of thyroid disease. Similarly, Shrum et al. [140] presented a suicide attempt by hanging that caused TS and cardiac arrest. Mathai et al. [117] reported a case of TS that was precipitated by injecting heroin into the neck. Kaplakam et al. described bedside tracheostomy as a possible trigger for TS [123].

Noncompliance with medications: The most common precipitating factor was noncompliance with medications in our analysis (32%). Galindo et al. [6] reported non-compliance in 15.4% of TS cases in comparison with 6% in thyrotoxicosis without storm. Moreover, insurance and side effects of ATDs interfere with the continuity of therapy [31,36,167].

Drug-induced TS: We found a few cases of amiodarone-induced TS that led to cardiomyopathy and CS [37,43,173]. Amiodarone is a class III antiarrhythmic drug with iodine content used to treat tachyarrhythmia, in around 14–18% of patients; it may result in amiodarone-induced thyrotoxicosis (AIT) [244]. AIT can occur in up to 10% of patients [244]. The risk of developing AIT can be seen after 18 months to 3 years; even so, it may occur after withdrawal, as the drug can remain in the tissues for a prolonged time. There are two types of AIT; type 1 generally occurs in patients with clinical thyroid disease [174]. According to our review, [43,93,103,174,212,229] six cases were described with AIT, including both types 1 and 2. Ozcan et al. presented a patient who was administered amiodarone infusion to control AF; however, the ventricular rate was not controlled, and arrhythmia worsened even after discontinuation of amiodarone and ICD insertion. Blood tests confirmed TS, presumably caused by amiodarone [174].

Moreover, AlShehri et al. reported a case of alemtuzumab (ALZ)-induced TS further complicated by patient noncompliance with ATDs for Graves’ disease due to multiple factors [78]. Mathew et al. presented a case of a body builder who suffered TS and myopericarditis leading to acute decompensated heart failure due to ingesting supplements with liothyronine and testosterone [171]. Exogenous testosterone decreases thyroid-binding globulin levels, resulting in enhanced levels of free thyroid hormones. Moreover, a case of tetrodotoxin intoxication due to ingestion of a seafood stew containing marine neurotoxins was described by Noh et al. [47]. Consequently, the patient was diagnosed with TS and reversible cardiomyopathy.

Kinoshita et al. [31] reported a case of TS-induced TCM in an elderly patient with diabetes mellitus, possibly triggered by empagliflozin, a sodium-glucose co-transporter 2 inhibitor (SGLT-2) that typically does not lead to TCM or TS. However, this patient had a history of thyroid disease, and the SGLT-2 inhibitor precipitated the TS.

Three cases reported the occurrence of TS following administration of iodine-based contrast [72,99,146]. Another case was reported by Kauth et al., in which a patient with Graves’ disease (not compliant with ATDs) developed iodine-contrast-induced TS. The patient was placed on IABP, TPE, and ECMO for refractory shock [184].

Acute illness and infection: The presence of infection can aggravate thyrotoxicosis in patients with TS, especially if unresolved or missed. Acute illnesses such as myocardial infarction, stroke, diabetic ketoacidosis (DKA), or hypoglycemia can lead to TS [31,46,104,126,133,165,169]. Das et al. [32] reported a case of TS with acute decompensated heart failure, possibly triggered by SARS-CoV-2 infection, in a 16-year-old female. This patient had pre-existing Graves’ disease and dilated cardiomyopathy; however, it was stable prior to COVID-19 infection. The overaction of the T helper cell response and elevation in interleukin-6 due to COVID-19 infection resulted in a change in thyroid gland functionality [32]. Prasankati et al. [105] described TS and COVID-19 in a patient without a history of heart or thyroid disease who presented with SVT.

White et al. reported a possible undescribed thyrotoxic crisis in which the patient’s initial presentation was pulseless electrical activity lasting for 15 min. There was significant LV dysfunction with an EF of 10% [77]. The myocardial biopsy revealed viral thyroiditis. The management involved ECMO support for 10 days, followed by levothyroxine administration, and the patient eventually survived.

**Cardiac manifestations of TS**: Thyroid hormone receptors are distributed in the myocardium and vascular tissue and can cause endothelial dysfunction and myocardial systolic and diastolic dysfunction [245]. The heart mainly relies on T3 hormones to regulate its activity because of the lack of significant intracellular ideiodinase activity in cardiomyocytes. This effect occurs through positive or negative regulation of the expression of key genes [246]. T3 exerts its effect on the heart via both genomic and non-genomic mechanisms and regulates cardiac function and cardiovascular hemodynamics through three main processes affecting hemodynamics (peripheral circulation), myocardial contractility, and heart rate. The presence of increased T3 levels would cause upregulation of positive cardiac gene expression and suppression of negative cardiac gene expression; positive gene expression includes upregulation of alpha-myosin heavy chain activity, sarcoplasmic reticulum Ca^2+^ ATPase and beta1-adrenergic receptors [246]. Hemodynamically, thyroid hormones decrease peripheral resistance in the arterioles through a direct effect on vascular smooth muscle cells and decrease mean arterial pressure. Such changes cause an increase in the heart rate and a decrease in the diastolic pressure. Vasodilation activates the renin–angiotensin–aldosterone system and increases Na+ absorption. In addition, T3 caused an increase in the red cell mass. All these changes lead to an increase in blood volume and preload and eventually cause an increase in cardiac output (ranging from 50 to 300% in patients with hyperthyroidism) [246]. Moreover, hypertension may occur because of the inability of the vascular system to accommodate the increase in stroke volume owing to a dramatic decrease in systemic vascular resistance (up to 70%) [246].

**Tachyarrhythmia**: Arrhythmias are frequently observed [167]. Sinus tachycardia is the most evident rhythm as it is a cardinal feature of TS [241,246]. Heart rate ≥ 150 bpm is associated with increased mortality in patients with TS [247]. A heart rate above 130 bpm has been reported in three-quarters of patients with thyrotoxic crisis [2]. However, AF is the most common arrhythmia in thyrotoxicosis, with a prevalence of 15%, and is typically of the persistent type rather than paroxysmal [179,248]. This was due to the pathological acceleration of diastolic depolarization of the sinoatrial node caused by the shortened action potential. Generally, thyroid hormones affect the cardiac conduction system, shorten the action potential, and increase the refractory duration of atrioventricular cells, thereby causing AF [246]. Moreover, overt sympathetic activity and an increase in thyroid hormone levels lead to excessive chronotropic and inotropic effects that contribute to tachyarrhythmia and myocardial ischemia [59]. Thyroid hormones augment beta-adrenergic receptor sensitivity to catecholamines and myocardial excitability. ECG findings such as atrial flutter, atrial tachycardia, ventricular tachycardia, and multifocal tachycardia have also been reported in TS. Waqar et al. reported that the proportions of AF, atrial flutter, VT, and SVT were 46%, 7%, 5%, and 1.5%, respectively [241].

In TS, AF is the most prevalent arrhythmia by 30–40% [111]. In a Japanese study, almost half of the TS patients who died had AF [3]. Evidently, there is a link between increased T4 serum level and AF incidence [248]. The incidence of AF has been linked to the increased sensitivity of myocytes to thyroid hormones as a result of high beta-adrenoreceptors on the surface of cardiac structures. This occurs because of the increased positive expression of beta-adrenoreceptor genes in response to increased T3 [245,246]. Rapid AF triggers hemodynamic collapse in the absence of an atrial kick, atrioventricular synchrony, and heart rate control [111]. Therefore, rapid AF is a precipitant of decompensated heart failure in TS [189]. Treatment of AF includes BBs, Class Ia and Ic antiarrhythmic agents, anticoagulation, digoxin, cardioversion (after exclusion of atrial thrombus), and amiodarone.

TS-induced ventricular fibrillation (VF)/sudden cardiac death (SCD): A few cases of TS-induced SCD have been described in the literature [70,181,182,224]. The rate of death due to cardiac arrest increases significantly within a short timeframe in patients with TS, particularly in the presence of coexisting coronary artery disease [249]. Joa et al. [65] reported TS 6 months after discontinuation of antithyroid medication in a female patient. On arrival, the initial presentation was VT followed by VF. Circulatory collapse in TS is multifactorial and has been described after the administration of BBs, calcium channel blockers (CCBs), persistent arrhythmias (AF or VT), severe decompensated HF, CS, or severe coronary spasm. Moreover, severe respiratory distress may contribute to respiratory muscle asthenia, severe pulmonary hypertension, and respiratory failure [250]. Even in euthyroid patients, a study showed that higher levels of free T4 were associated with an increased risk of SCD, with a hazard ratio of 1.87 per 1 ng/dL increase in free T4 [250]. Therefore, we recommend careful follow-up of patients with increased free T4 levels.

J-point elevation on ECG in TS: Although early repolarization (elevation of the QRS–ST junction (i.e., J-point elevation)) in at least two electrocardiographic leads is a common benign finding, it can be an arrhythmogenic substrate leading to VF [179]. J-point elevation represents a transmural voltage gradient between the endocardium and epicardium during ventricular electrical activation. In patients with TS, BBs may be an unsafe choice in the presence of early repolarization on ECG as it augments the elevation of the J-point and ST segment, increases the voltage gradient between the endocardium and epicardium, and enhances arrhythmogenicity. Ueno et al. [181] reported that a 69-year-old male presented with TS, and his ECG showed sinus rhythm with J-point elevation in the inferior and anterior leads. The patient was treated with landiolol (short-acting BB). During hospitalization, the patient developed AF followed by persistent VF, necessitating percutaneous cardiopulmonary support. Despite cardiac rhythm correction, the patient died a few days later due to multiorgan failure.

**Troponin release, Myocardial Injury, and Infarction (AMI)**: The actual incidence of AMI in patients with TS is not well known or is under-defined. Myocardial injury with or without ECG changes has been reported in most cases of TS and is defined as a cardiac troponin T concentration of 0.03 ng/mL or more or high-sensitivity troponin T concentration of 10 ng/L or more for women or 15 ng/L or more for men. AMI of either type I or II may occur in patients with thyrotoxicosis, with or without atherosclerosis, and most articles did not define the significance of elevated troponin levels in TS, especially when coronary angiography was not performed. Most patients with AMI are treated conservatively without thrombolytics or intervention because the majority has non-obstructive or non-atherosclerotic causes. In addition, multiorgan dysfunction and hemodynamic instability are limiting factors for aggressive treatment of AMI. Moreover, a slight increase in troponin levels in TS may be due to tachycardia, coronary artery spasm, or Takotsubo cardiomyopathy [241]. In a small sample size case series (*n* = 5), TS was initially misdiagnosed as AMI and CHF, which delayed on-time treatment of the storm [59].

**TS-induced coronary spasm**: Non-atherosclerotic causes of myocardial injury include an imbalance (mismatch) between oxygen supply and demand, which can occur with epicardial coronary artery spasm, tachyarrhythmias, or thyrotoxicosis. These factors can lead to type 2 acute MI [251]. Excess Thyroxine is associated with coronary spasms in young patients [252,253]. Omar et al. [96] described a 40-year-old male presented with an out-of-hospital cardiac arrest and AMI. Coronary angiography revealed spasm in the right coronary artery. The patient was successfully treated with intracoronary nitrate, BBs, CCBs, and intravenous fluids. Factors explaining excess thyroxine-induced coronary spasm include an increase in cellular calcium content, sympathetic activity, adrenergic receptor sensitivity, increased receptor numbers, hyper-reactivity of vascular smooth muscles, and coronary vasomotor tone abnormalities [181,251,253,254,255].

Zheng et al. [253] reported the case of a young man with inferior myocardial infarction. Laboratory tests revealed new evidence of toxic thyroiditis and patent corona. The authors concluded that it was thyrotoxicosis-induced coronary spasm; however, we did not include it in our cases, as they did not document the evidence of a storm. The same authors reviewed 21 cases of thyrotoxicosis-induced AMI between 2000 and 2014 and found normal coronary arteries in 13 cases; TS was described in 1 case [64,253]. Moreover, three cases showed clear evidence of coronary vasospasm.

**Acute Heart failure and cardiomyopathy**: Heart failure may be the initial presentation of TS and the main cause of mortality [10]. High-output heart failure is often observed in patients with TS, owing to the overabundance of thyroid hormones. This type of heart failure includes cardiac output elevation compared with metabolic demand and a decrease in systemic vascular resistance mediated by the peripheral vasodilator adrenomedullin [10,162,177]. Consequently, it may progress to dilated cardiomyopathy and dysrhythmias [23]. Most patients have LVF or biventricular failure; however, isolated right-sided heart failure secondary to pulmonary artery hypertension can occur [256].

Dilated Cardiomyopathy (DCM): Dilated cardiomyopathy is characterized by progressive heart muscle disease and is the most common cardiomyopathy phenotype. Dilated thyrotoxic cardiomyopathy, an unusual TS phenotype, initially manifests in 6% of patients; however, severe LV dysfunction is observed in <1% of patients [1,30]. The presence of unregulated persistent tachycardia in patients with hyperthyroidism may precipitate DCM in 6–15% [53]. It is crucial to recognize DCM as a potentially reversible and unusual manifestation in patients with thyrotoxicosis patient [241]. Excess thyroxine centrally stimulates activity in the sympathetic nervous system by positively regulating β1-adrenergic receptors and upregulating sarcoplasmic reticulum Ca^2+^ ATPase, which is involved in excitation-contraction coupling and calcium-induced calcium release [257]. Calcium released from the ryanodine receptor in the sarcoplasmic reticulum activates the myocardial myofilaments, leading to positive inotropy [257].

Free T3 and T4 increase the expression of the more rapid contractile isoforms of the alpha-myosin heavy chain. In addition, these hormones stimulate erythropoietin secretion, contributing to increased blood volume, leading to high-output CHF. T3 leads to an increase in stroke volume and pulse rate and promotes peripheral vasodilation, causing a decrease in systemic vascular resistance, which in turn activates the renin–angiotensin system, leading to fluid and salt retention [30]. Notably, persistent tachycardia impairs myocardial contraction as the activity of the Na/K-ATPase pump declines, in addition to the downregulation of beta-adrenergic receptors [1]. These factors eventually lead to DCM during thyrotoxicosis.

According to our review, 11 cases of TCMP have been previously reported. Takotsubo cardiomyopathy is seen in 17.7% of thyroid diseases, of which 5.9% are attributed to hyperthyroidism [31]. TCMP is also defined as stress cardiomyopathy and broken heart syndrome that can lead to transient or reversible heart failure [46,50]. An increase in thyroid hormone levels above the normal range leads to intensified adrenergic activity, resulting in stress-induced cardiomyopathy [28,46].

Brain natriuretic peptide (BNB) in TS: Dyspneic patients with BNP > 100 pg/mL or NT-proBNP > 400 pg/mL are more likely to have CHF [258]. In this review, there were several cases of TS with high BNP levels (56 cases). Arai et al. [19] reported a case series of eight TS patients with on-admission BNP values of >700 pg/mL requiring IABP (*n* = 4), and one of them had IABP, ECMO, and CRRT. The LVEFs ranged between 14 and 20%, and the BWS scores ranged from 80 to 90, and all survived after a long hospital stay.

**Pericardial effusion**: There were seven cases of pericardial effusion and TS [33,41,106,163,193,205]. Pericardial effusion and pericarditis are rare in thyrotoxic diseases and may resolve without intervention after improvement of hyperthyroid status. Bui et al. reported pericardial effusions in 7 out of 12 patients with thyrotoxicosis due to Graves’ disease [259].

**Circulatory collapse and Cardiogenic shock**: CS is an infrequent complication of TS, with a high fatality rate [19]. In patients with impaired systolic function, CS has a mortality rate of up to 30% [167]. A Japanese study showed that the presence of shock increases the likelihood of mortality four-fold [3]. However, information regarding CS in TS is limited. Some factors are associated with the occurrence of thyrotoxic-induced CS, such as pre-existing CHF, valvular heart disease, AF, coagulopathy, and hepatic, renal, and pulmonary dysfunction [41]. Risk factor identification is crucial because these patients would require more intensive care and caution [41]. Additionally, drugs that may precipitate CS as BBs and CCBs must be discontinued to avoid further deterioration. According to Mohananey et al., approximately 51.8% of patients presenting with AF suffer from CS, and 54.8% have consequently died [260]. In addition, there was an increase in the incidence of CS from 0.5% in 2003 to 3% in 2011, and a decrease in mortality from 60.5% in 2003 to 20.9% in 2011 in a study from the USA [260]. Iwańczuk [64] reported the case of a 51-year-old female presented with TS, including multiorgan dysfunction and AMI. Her coronary arteries were patent, and because of catecholamine-resistant shock, she was administered dopamine, dobutamine, norepinephrine, and IABP. She underwent subtotal thyroidectomy after a week and survived without any disabilities.

**Pulmonary Embolism (PE)**: Around 10–40% of hyperthyroidism cases are found to have arterial embolism [230]. Notably, hypercoagulability is associated with hyperthyroidism. These cases show a link between thyrotoxicosis/TS, cerebral thrombosis, deep vein thrombosis, and PE [142,230]. A variety of factors could play a role in this regard, such as an increase in factor VIII activity, Von Willebrand factor, and tissue plasminogen activator inhibitor-1. Lin et al. concluded in a 5-year follow-up study that patients with hyperthyroidism had a 2.3-times higher risk of developing pulmonary embolism [261].


**Modalities of treatment of TS with myocardial involvement**


A multidisciplinary approach must be applied to patients with TS, because multiple organs are affected. It is crucial to adequately evaluate and avoid misdiagnosis of TS as early management plays a crucial role in the outcome. Generally, the Japanese guidelines for treatment include treatment of thyrotoxicosis (reduction in thyroid hormone secretion and production), management of systemic manifestations (fever, dehydration, shock), organ-specific manifestations (cardiovascular, neurological, and hepatic-gastrointestinal), triggers, and definitive therapy [247].

(A)Pharmacological treatment modalities

Initial Management: According to our findings, 90% of patients were administered ATDs once diagnosed with TS. This classic regimen includes thioamides such as carbimazole (CBZ), methimazole (MMI) and propylthiouracil (PTU). They act by inhibiting thyroid peroxidase (TPO), thereby blocking the synthesis of T3 and T4 from thyroglobulin [17]. MMI and CBZ have higher potency than PTU. CBZ/MMI has a longer half-life and prolonged time of action; therefore, it may be administered as one dose daily rather than several times daily. However, PTU tends to be more effective in the treatment of TS as it also blocks the conversion of peripheral deiodinase-mediated T4 to T3 [17]. However, Matsubara et al. [167] reported a case of agranulocytosis following MMI administration, which required thyroidectomy as the definitive treatment. Voll et al. [36] described the use of VA ECMO, Lugol’s iodine, and thyroidectomy to treat TS in a patient who developed neutropenia due to CBZ.

Table 4 shows the strength of the recommendation and quality of evidence for the measures of TS treatment [247]. In Graves’ disease, ATDs should be initiated as soon as possible, and adherence to treatment is vital. Large doses of inorganic iodide, simultaneously with ATDs, should be administered to treat Graves’ disease complicated with TS.

Steroid: Corticosteroids should be administered as prophylaxis for relative adrenal insufficiency caused by the hypermetabolic state in TS. Large doses of steroids have been shown to inhibit thyroid hormone synthesis and peripheral conversion of T4 to T3 [247]. Corticosteroids can be administered at a dose of 300 mg/day of hydrocortisone or 8 mg/day of dexamethasone. It controls adrenal insufficiency post-TS.

Beta-Blockers: BBs are essential components of the standard treatment of TS. BBs counteract the indirect effects of thyroid hormones by decreasing systemic vascular resistance, increasing cardiac output by 300%, and allowing peripheral vasodilatation [1]. The documented cases utilized non-cardioselective BBs (NCBBs) and cardioselective BBs (CBBs). The reported mechanisms of action, standard doses, indications, and notable adverse effects are shown in Table 5. The life-threatening side effects include CS, acute heart failure, and bradycardia. They are administered via multiple routes and at varying doses for different durations. Propranolol is a favorable choice for various reasons; notably, it has the additive effect of inhibiting the conversion of peripheral latent T4 to T3 [175,177]. Due to short-term effects, landiolol, esmolol (intravenous), or bisoprolol (oral)) was selected as the first choice, and discontinuation of BBs should not be considered when the heart rate is <80 bpm and systolic blood pressure is <80 mmHg. If bronchial asthma is suspected, the patient is switched to verapamil or diltiazem [247].

Beta-blocker associated with circulatory collapse: BBs therapy has been described as a double-edged sword in TS [161]. Patients with prior clinical thyrotoxic cardiomyopathy or subclinical disease, especially in a setting of low-output heart failure, are prone to circulatory collapse when administered propranolol [175,177]. This may be attributed to NCBBs averting the compensatory hyperadrenergic state caused by thyrotoxicosis, and thus, a sharp decrease in the cardiac output in events as a TS leads to circulatory collapse [175]. Approximately 25.8% of cases reported hemodynamic instability and circulatory collapse, possibly due to BBs. In these cases, some were administered concomitantly with BBs and CCBs. Both agents are known to have negative inotropic effects [40]. Propranolol was the BB used in 46% of cases. Patients administered propranolol or atenolol tend to require extensive resuscitation [52]. Evidently, in conditions with signs of heart failure and low ejection fraction during thyrotoxic crisis, other agents are recommended [10,20,46,47]. Such agents are cardioselective BBs with shorting-acting properties, such as landiolol and esmolol, because titration and cessation are attainable [10,26,51,52]. Moreover, Voll et al. highlighted that even when dobutamine was administered during circulatory collapse alongside a high dose of propranolol, it was deemed less effective [36]. However, Misumi et al. reported the first case in which landiolol was responsible for cardiac arrest in TS [51]. Lim et al. reported a patient suffering from thyrotoxic crisis necessitating an esmolol infusion to manage his tachycardia; however, as a result, he went into CS shortly after the infusion [161]. Nonetheless, the use of short-acting BBs is advised; however, upon the onset of decompensated heart failure they become harder to tolerate [40]. Yamashita et al. reported the successful use of landiolol instead of bisoprolol in a patient with TS and heart failure complicated by AF [26]. It is worth noting that approximately 55% of cases did not have hemodynamic instability, circulatory collapse, CS, or arrest after the administration of BBs. It is important to highlight that some patients may present with circulatory collapse or cardiac arrest due to the pathophysiology and nature of TS itself, as seen in 29% of the cases.

Calcium channel blockers (CCBs): Few authors have reported the use of CCB for the treatment of TS. However, Saakan et al. attributed diltiazem to the reversible CS in the setting of TS, as CCBs are atrioventricular blocker agents, leading to worsening of cardiac output and hemodynamic instability in a few cases [15].

Ivabradine: Hyperpolarization-activated cyclic nucleotide-gated (HCN) channel blockers, such as Ivabradine, act by selectively and specifically inhibiting the cardiac pacemaker current (I_f_) of the sinoatrial node and do not affect contractility. Frenkel et al. described a 37-year-old patient with thyrotoxicosis and CHF [62]. In attempts to regulate the heart rate using propranolol at high doses, it was deemed ineffective. The patient was prescribed Ivabradine, a, and within 48 h, the heart rate was well controlled [62].

Amiodarone: It can be used in patients with TS for the treatment of tachyarrhythmia such as AF, flutter, and paroxysmal tachycardias [244]. Owing to its iodine content, it leads to thyroid dysfunction, as it can increase iodine 50–100-fold. Around 14–18% of patients using amiodarone regularly develop AIT [174,244]. Typically, a dose of 200–400 mg daily is safe and maintains TSH serum levels; however, if administered intravenously, it may increase them [244]. It is considered effective in patients with AF, thyrotoxic cardiomyopathy, and CS after receiving necessary ATDs [40].

Although amiodarone can cause AIT, Yamamoto et al. reported a patient with an initial diagnosis of multifocal atrial tachycardia complicated by CS [1]. Owing to unresolved refractory tachyarrhythmia, the decision was made to insert an IABP and continuous amiodarone infusion [1]. As the patient’s condition improved, it was later found in blood samples taken from the first (pre-amiodarone) to the seventh day that the patient had TS [1]. The author declared amiodarone to have been curative for TS even though she did not receive the classic TS treatment regimen [1]. Because amiodarone has high iodine levels, it can block thyroid cells from iodine uptake.

Anti-failure treatment: CHF should be treated early, and serial echo and BNP are advisable in addition to Swan-Ganz catheter insertion in severe cases and to control AF. The following can be used: digoxin, furosemide (intravenous), nitrate (sublingual or intravenous), and/or carperitide (intravenous α-human A-type natriuretic peptide); noninvasive positive pressure ventilation (NIPPV); vasoconstrictor agents; and cardiotonic agents [247,262].

Inotropes and vasopressors: In cardiogenic or refractory shock, dopamine should be administered intravenously at a dose of 5–20 μg/kg/min when systolic blood pressure is between 70 and 90 mmHg. Dobutamine at a dose of ~10 μg/kg/min should be considered when the patient is in CS and systolic blood pressure is ≤70 mmHg. Norepinephrine at a dose of 0.03– 0.3 μg/kg/min is also used when the patient’s hemodynamic condition does not improve with these agents, or systolic blood pressure is ≤70 mmHg [247]. Mechanical support systems should be considered early, or if these medical measures fail or are contraindicated.

(B)Non-pharmacological therapy

**Mechanical treatment of TS**: The three main extracorporeal support systems (ECMO, TPE, and CRRT) can be useful as bridges for stability and definitive surgery in TS. Both ECMO and TPE were used concurrently in multiple cases, and only a few cases utilized all three. Evidence for the use of these systems arises from case reports or a few case series, as there are no prospective clinical trials. The use of these systems is costly and does not have side effects. Therefore, they require an expert team and appropriate patient selection and timing.

(a)Extracorporeal Membrane Oxygenation (VA-ECMO): In 2021, Lim et al. [161] reported that there were 27 cases in the literature at the time of thyrotoxic crisis requiring ECMO, and 85% of these patients survived. In severe cases, first-line pharmacotherapy may not be sufficient to restore cardiovascular function to normal levels after TS development. When faced with this, extracorporeal modalities are implemented. Among the 256 cases, the use of ECMO was reported in 16.3% of cases; hence, it was the most used mechanical support. ECMO bypasses the heart and lungs and provides gas exchange through the external membrane [161]. This process supports the heart by temporarily relieving the heart of its functions to allow it to heal, while thyroid hormones normalize, and the euthyroid state is restored [263].

ECMO is mainly indicated in cases of acute severe cardiac or pulmonary failure unresponsive to conventional therapy (ELSO guidelines) [264]. In addition, it has been reported in multiple cases that the use of ECMO contributes to more successful outcomes until thyroidectomy is performed because of the stabilization of myocardial function [161]. Furthermore, when used in concordance with other extracorporeal modalities, the results may be enhanced, and treatment becomes more effective [161,263].Despite the recent incorporation of ECMO in the management of endocrine emergencies, the overall survival rate could be unreliable because of possible publication bias, patient selection bias, and small sample size [161]. The cause of death might not directly correlate with the use of ECMO but rather with the severity of cardiomyopathy, shock, and other complications [42,161]. Common complications of ECMO include bleeding, thrombosis, limb ischemia, and stroke [77,161].

(b)Therapeutic plasma exchange (TPE): it is a class II indication of TS. TPE is one of the most effective methods for eliminating excess thyroid hormones circulating in patients with TS [9]. It uses a purification technique that rapidly filters out large molecular substances from the plasma, reducing protein-bound and free T4 and T3 [161]. According to the American Society for Apheresis, TPE is a category III indication for TS and its use is based on individual cases [161]. It is important to note that during this process, clotting factors and immunoglobulins may also be filtered out; therefore, the patient should be infused with replacement colloid and blood products to avoid the risk of bleeding and infection [163]. TPE should be implemented early in the treatment course of TS to ensure the best results [163]; however, in some cases, it may be delayed owing to more pressing complications, which can cause technical difficulties, such as the need for CVVHD implementation for acute kidney injury and metabolic acidosis [48]. TPE may also be incorporated into multimodal therapeutic course [161]. TPE can be used in combination with ECMO, Impella, or CRRT [38,161]. TPE is also used as a bridging treatment until the patient becomes sufficiently stable to undergo thyroidectomy [163]. TPE can reduce all free and total thyroid hormones by 10 to 80%, reduce autoantibodies and cytokines, and remove 5-monodeiodinase to inhibit the conversion of T4 to T3 [161]. Multiple cycles of TPE are required in some cases, as thyroxine distribution is mainly in the extrahepatic tissue (34%), intravascular (26%), extracellular fluid (26%), and liver (14%) [161,265].(c)Continuous Renal Replacement therapy (CRRT) and continuous venovenous hemofiltration (CVVH): CRRT includes the use of large volumes of room-temperature fluids (dialysate and replacement fluids), which can cause hypothermia. In addition, intravenous infusion of albumin and plasma in CRRT increases the ability of proteins to bind free thyroid hormones [38]. CRRT is a treatment method that utilizes intermittent hemodialysis and peritoneal dialysis [266]. It has been used in patients with acute kidney injury (AKI) who are hemodynamically unstable secondary to TS. (d)CVVH is one of the modes of CRRT, which was described in six of this review cases. Our data showed that CRRT was significantly associated with a high mortality rate, particularly in patients with acute renal failure. CVVH uses convective clearance to remove toxins and solutes from the patient’s circulation, whereas CVVHD relies on diffusive clearance to remove the same toxins and solutes [266]. CVVH helps prevent sequelae resulting from metabolic and hemodynamic instability.(e)Intra-aortic balloon pump (IABP): This device is used in cases of acute heart failure with shock after ineffective inotropic or vasopressor administration. The IABP works by inflating the balloon during diastole and aortic valve closure and rapidly deflating before systole [267]. This results in a reduced afterload, which consequently improves cardiac output by increasing stroke volume and ejection fraction. Around nine percent of the cases used IABP, and a few patients died immediately thereafter. IABP was used alongside ECMO (nine cases used ECMO + IABP) as a means of circulatory support, whereas the underlying cause of heart failure was treated (TS). Some authors prefer ECMO for IABP [162].(f)Impella: The Impella device pumps blood from the left ventricle into the ascending aorta and helps maintain systemic circulation at an upper rate between 2.5 and 5.0 L/min. The use of the Impella over ECMO is based on the concept of ventricular unloading to allow ventricular time to recover. It is a very small catheter-based device used as ventricular support in patients with CHF and CS [268]. Impella (Bi-pella) was used in three of our cases. In one case, the patient was initiated on esmolol drip, but deteriorated immediately after signs of biventricular failure. He was placed on the CP Impella (LV), but inadequate improvement called for the use of RP Impella (RV) [168]. The use of biventricular Impella allowed the treatment of underlying heart failure and yielded significantly better outcomes.

**Hypothermia resuscitation**: Fu et al. described an in-hospital TS-induced cardiac arrest in a 24-year-old woman. After resuscitation, the patient was hyperthermic (39 °C) and comatose [61]. Targeted temperature management with intravascular cooling is initiated to prevent neurological damage. After 72 h, the target temperature of 34 °C was reached; however, the patient experienced intermittent fever. Thus, an ice blanket and an ice cap were used until the patient stabilized. Shimoda et al. also used a systematic cooling approach with a cooling blanket in TS patients who presented with heart failure, atrial flutter, fever, liver failure, DIC, and elevated soluble interleukin-2 receptor (sIL-2R) levels [192].

**Radioactive iodine ablation (RAI)**: RAI is considered a safe and effective option for destroying thyroid tissue in many thyroid disorders such as cancer and hyperthyroidism. In our review, RAI was used in four cases. All patients presented with Graves’ disease and secondary heart failure that were successfully managed. RAI was performed after successful management to minimize the risk of relapse. All the patients developed hypothyroidism after RAI [23].

**Surgical treatment**: Thyroidectomy entails removal of the thyroid gland to eliminate the primary source of excess secretion of thyroid hormones [269]. Although it is the last line of treatment for TS, it remains the most effective option. Often, when the patient’s TS is refractory to medications and does not improve, surgical intervention is deemed necessary. Notably, thyroidectomy does not ensure a future relapse [36]. Our analysis revealed a significant association with lower mortality rate. However, frailty in patients with refractory TS makes surgery difficult in the early phase of presentation [24].

Table 6 shows the indications and complications of non-pharmacological treatment.

**Outcome and Prognosis**: TS represented 16% of hospitalized patients with thyrotoxicosis and had 12 times the mortality rate compared with thyrotoxicosis without a storm [6]. Another report showed that hospital mortality of TS can reach 10–75% [270]. In the review, the mortality rate was found to be 13.5%. Twenty patients developed circulatory collapse and hemodynamic decompensation after use of BBs and/or calcium channel blockers. Ten patients experienced complications due to MOF or sepsis regardless of the mechanical treatment provided. Two patients who did not receive adequate medications upon arrival sustained acute heart failure and could not be treated. Moreover, one patient who was non-compliant with ATDs succumbed to the second admission due to another TS encounter [19]. Two patients experienced acute respiratory distress despite medical efforts and expired [46,121]. One patient died of MOF within hours of resuscitation of three cardiac arrests after complete heart block [239]. Chao et al. reported the death of two male patients: one patient died due to MOF while on ECMO and one due to CS [59]. Another patient died due to cardiomyopathy, MOF, and DIC even though they were on ECMO, CRRT, and TPE, and underwent LVAD insertion [73]. CRRT is associated with high mortality in patients with acute renal injury [271]. The common causes of death after TS are MOF in one-quarter of cases, CHF in one-fifth of cases, and arrhythmia in 8% based on one Japanese study; this study on multiple logistic regression analysis showed that the presence of MOF increases the likelihood of death 10-fold [3]. Moreover, a markedly elevated sIL-2R level was reported as a potential novel pathogenic factor in the development of TS complicated by MOF as it activates the systemic immune response [191]. However, this was observed in a case report, and a larger study is needed to support this finding.

Based on this systematic scoping review, we proposed an algorithm for the management of TS with cardiac involvement (Figure 4).

**LIMITATIONS**: This systematic scoping review has some limitations, including papers published in languages other than English and gray literature. Moreover, articles that presented cases without specified treatment modalities, abstracts, or abstract posters (unless the required information was available) were not included. We did not use diagnostic criteria other than BWS, such as the Japanese criteria [3], which may have underscored the number of cases in this review. However, the Japanese study did not examine myocardial involvement and its treatment modality, and its use was limited to a few countries. The quality of the studies was not assessed as the majority were case reports. This work draws the attention of readers for the difference between the systematic scoping review and the well-known systematic review. The scoping one is exploratory, addresses a broad question to explore the extent of the available evidence, organizes it, pinpoints gaps, and determines whether it would be worthwhile to conduct a systematic review [272].

## 5. Conclusions

The exact mechanism underlying the development of TS in uncomplicated thyrotoxicosis is not yet well defined; however, most manifestations of the latter occur in an exaggerated and wider manner during the storm. This could be due to the abrupt increase and availability of free thyroid hormones in the circulation, in addition to the enhanced response of the tissue receptors (which increase in number) to the hormone and catecholamine surge. Therefore, early and appropriate treatment of severe thyrotoxicosis is crucial to prevent the progression to TS and its higher fatalities. Frontline physicians should be aware of “TS” and not only “thyrotoxicosis” and the on-time appropriate treatment. The index of suspicion should be high, especially in the absence of a prior history of hyperthyroidism or clear triggers, as it may be missed in 30% of cases. Management should be guided by the affected end organ, indication versus contraindication (safety) of certain therapies, and the prevention of recurrence. The early diagnosis and management of TS in cardiac settings, including pharmacological, mechanical, and surgical modalities, may save high-risk patients. Mechanical support is required to bridge the gap between stability and definitive treatment. Sex matters in the presentation, treatment, and mortality of these populations, to a certain extent. However, further large-scale and well-designed studies are required.

## Figures and Tables

**Figure 1 diagnostics-13-03028-f001:**
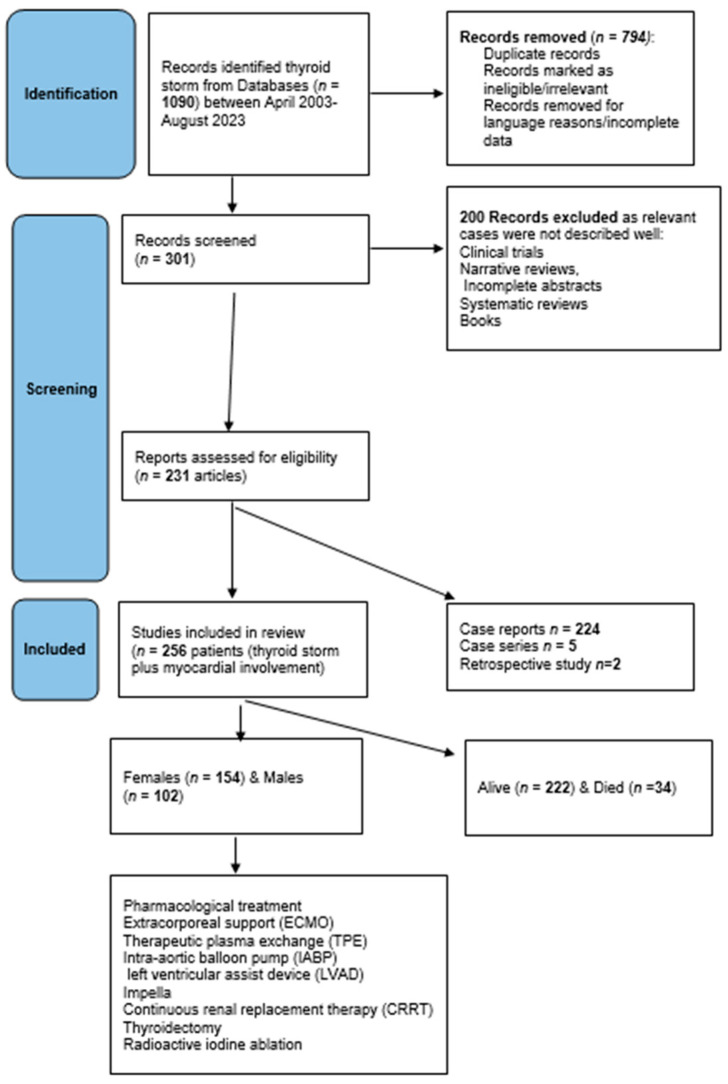
Flowchart of the systematic scoping review design.

**Figure 2 diagnostics-13-03028-f002:**
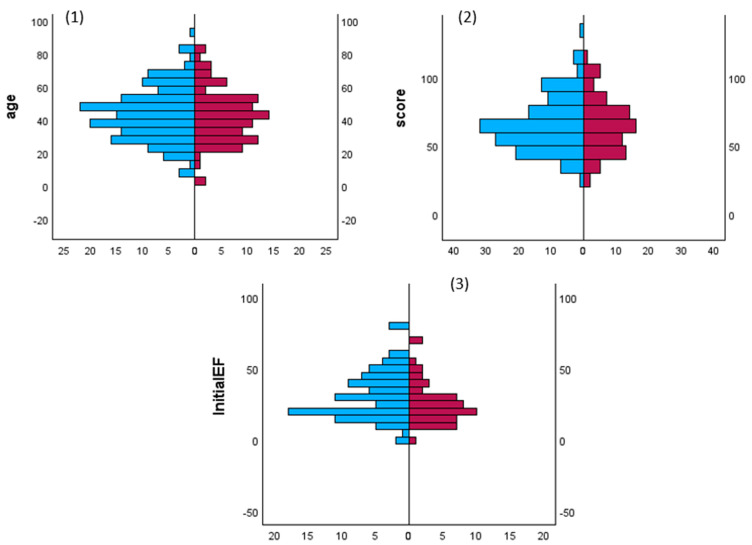
Comparisons of patients age, TS scores and initial LV ejection fraction (initialEF) in both patients’ sex. Mann–Whitney U test for frequency of gender and (1) age groups, (2) Burch–Wartofsky point scale points and (3) initial left ventricular ejection fraction in males (red) and females (blue) patients with thyroid storm.

**Figure 3 diagnostics-13-03028-f003:**
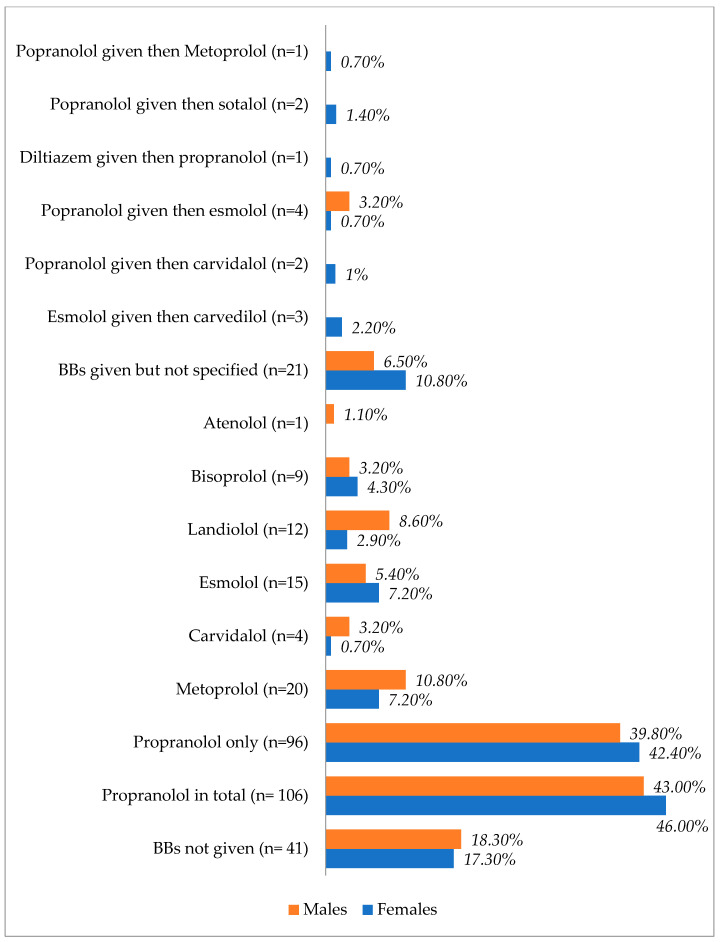
Types and proportions of used beta blockers based on the patients’ sex.

**Figure 4 diagnostics-13-03028-f004:**
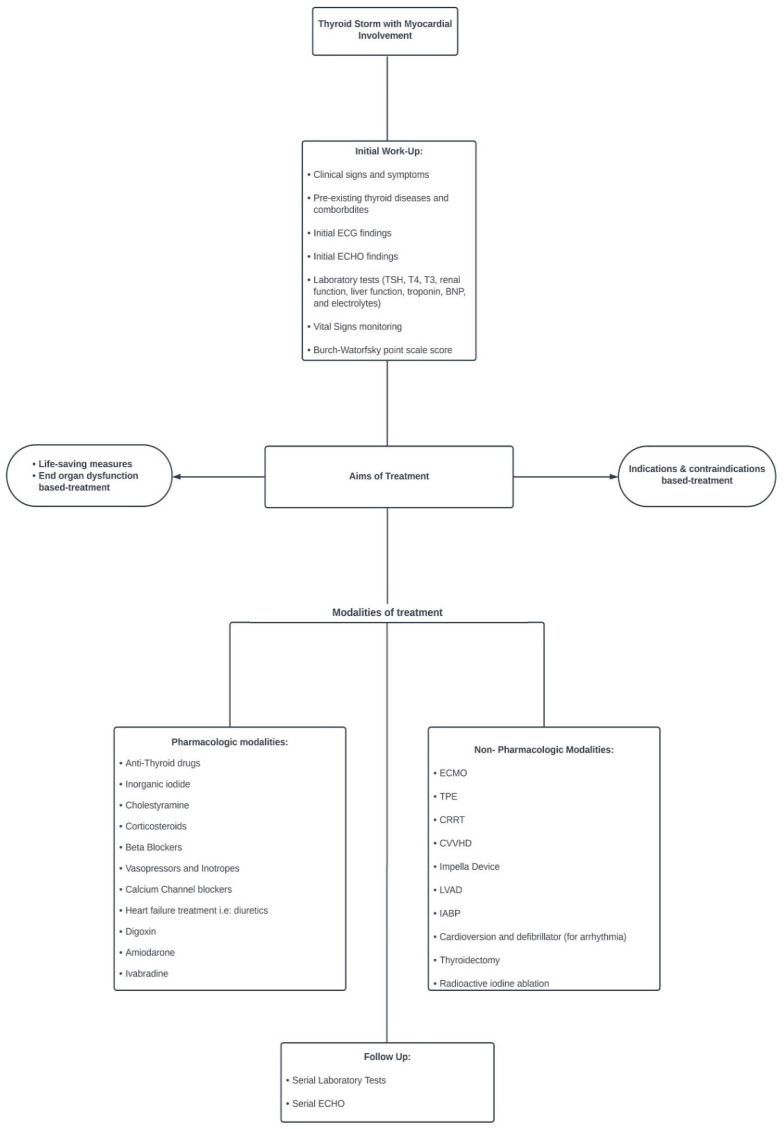
Algorithm for management of thyroid storm with cardiac involvement.

**Table 1 diagnostics-13-03028-t001:** Comparison between females and males based on the cardiac involvement.

Variable	Overall (*n* = 256)	Female (*n* = 154) 60%	Male (*n* = 102) 40%	*p* Value
Age; median and IQR	42.5 (31–58)	43 (31–52)	42 (29–53)	0.77
BWS point; median and IQR	60 (50–75)	60 (50–75)	60 (45–75)	0.67
Non-compliant to treatment	80/252 (32%)	31%	33%	0.56
High serum natriuretic peptide level	56/256 (22%)	26%	16%	0.05
Positive cardiac troponin	19/250 (7.6%)	12%	7%	0.48
Acute myocardial infarction	25/250 (10%)	8%	7%	0.48
Presented with cardiac arrest	14/246 (5.7%)	7.5%	3.0%	0.13
Initial LVEF%; median and IQR	25 (19–40)	30 (20–43)	23 (15–32)	0.05
Follow-up LVEF%	50 (36–58)	51 (43–59)	48 (35–57)	0.17
Acute heart failure (any grade) *	112/239 (47%)	41%	46.5%	0.02
Dilated cardiomyopathy	31/249 (12.4%)	11%	23%	0.01
Heart failure with preserved EF	5/233 (2.1%)	2.2%	2.0%	0.91
Pulmonary edema	49/236 (21%)	26%	13%	0.01
Takotsubo cardiomyopathy	11/234 (4.7%)	7.1%	1.1%	0.03
Pericardial effusion	7/254 (2.8%)	2.6%	3.0%	0.88
Cardiogenic shock	78/242 (32%)	28%	38%	0.27
Cardiac arrest	61/238 (25.6%)	23%	29%	0.23
Pulseless electrical activity	9/243 (3.7%)	2.7%	5.2%	0.33
Asystole	1/238 (0.4%)	0.7%	0.0%	0.42
Ventricular fibrillation	12/244 (4.9%)	4.1%	6.2%	0.48
Ventricular tachycardia	6/245 (2.4%)	2.7%	2.0%	0.72
Beta-blocker-induced collapse	63/244 (25.8%)	25%	27%	0.66
Atrial fibrillation	123/249 (49%)	44%	58%	0.02
Atrial flutter	16/245 (6.5%)	5.4%	8.2%	0.39
Sinus tachycardia	69/246 (28%)	35%	17%	0.002
Multifocal atrial tachycardia	3/245 (1.2%)	1.4%	1.0%	0.80
Supraventricular tachycardia	14/246 (5.7%)	7.5%	3.0%	0.13
**Multiorgan failure**	120/250 (48%)	47.4%	49%	0.80
Respiratory failure	39/227 (17.2%)	15.9%	19.1%	0.54
Renal failure	47/227 (20.7%)	37%%	49%	0.06
Liver failure	95/229 (41.5%)	47%	53%	0.03
Heart failure/cardiomyopathy	161/229 (70%)	74%	65%	0.14
**Mortality**	34/255 (13.5%)	11.8%	16.2%	0.33

LVEF: left ventricular ejection fraction, IQR: interquartile range, * Heart failure was considered if the author mentioned “heart failure” “pulmonary edema”, “CHF”, “diastolic HF”, or “systolic dysfunction” with or without reporting left ventricle ejection fraction percent or serum natriuretic peptide level as the latter two were not performed/reported in all cases.

**Table 2 diagnostics-13-03028-t002:** Comparison between females and males with thyroid storm and cardiac involvement.

Variable	Overall (*n* = 256)	Female (*n* = 154)	Male (*n* = 102)	*p* Value
**Pharmacological treatment**				
Amiodarone	19/235 (8.1%)	8.8%	6.5%	0.34
Inotropes/vasopressor	81/241 (33.6%)	28.6%	41.5%	0.08
Digoxin	25/241 (10.4%)	9.7%	11.5%	0.65
Steroids	172/247 (70%)	65.5%	75.8%	0.05
Calcium channel blockers	30/238 (12.6%)	11.8%	13.8%	0.64
Beta blockers	191/232 (82%)	60%	40%	0.84
Anti-thyroid drugs	228/251 (91%)	90%	92%	0.57
Ivabradine	2/253 (0.8%)	0.7%	1.0%	0.877
**Non-pharmacological treatment**				
CRRT/CVVHD/dialysis	34/245 (13.8%)	16%	8.6%	0.12
Implantable cardioverter-defibrillator	5/244 (2%)	0.7%	4.2%	0.05
VA-ECMO	40/246 (16.3%)	13.3%	21%	0.12
Left ventricular assist device	3/245 (1.2%)	0.7%	2.2%	0.31
Impella device	5/245 (2%)	1.3%	3.2%	0.32
Intra-aortic balloon pump	23/246 (9.3%)	8.0%	12.5%	0.29
Therapeutic plasma exchange (TPE)	40/246 (16.3%)	13.9%	20.1%	0.21
Radioactive Iodine ablation	8/245 (3.3%)	4.0%	2.1%	0.42
Thyroidectomy	46/247 (18.6%)	17.2%	21%	0.47

VA-ECMO: venoarterial extracorporeal membrane oxygenation, CRRT: continuous renal replacement therapy, CVVHD: continuous venovenous hemodialysis.

**Table 3 diagnostics-13-03028-t003:** Comparison between alive and deceased patients after thyroid storm with cardiac affection.

Variable	Alive (*n* = 217) 86.86%	Dead (*n* = 34) 13.5%	*p* Value
Age, median and IQR	42 (30–53)	43 (32–49)	0.91
BWS points median and IQR	60 (50–75)	65 (55–75)	0.78
Initial ejection fraction % median and IQR	27 (20–40)	20 (15–37)	0.37
Female gender	62%	53%	0.43
Male gender	38%	47%	0.43
Graves’ disease	29%	21%	0.41
Not known to have thyroid disease before admission	6.5%	12%	0.45
Non-compliant to ATD before admission	31%	37.5%	0.72
Atrial fibrillation after admission	52%	33.3%	0.06
Beta blocker (BB) use	82%	87%	0.49
Shock after admission	42%	55%	0.19
Pulmonary edema on admission	21%	17%	0.56
Takotsubo cardiomyopathy	4.0%	7.0%	0.50
Acute myocardial infarction	7.1%	9.1%	0.46
Acute liver failure	40%	56%	0.12
Acute renal failure	18.4%	41%	0.008
Ventricular fibrillation	3.8%	13.3%	0.02
Ventricular tachycardia	2.4%	3.3%	0.78
BB-induced circulatory collapse	21.3%	57.6%	0.001
Mechanical therapy (any)	47%	57%	0.30
ECMO	15%	28%	0.056
CRRT	5.0%	26%	0.001
Thyroidectomy	21%	6.5%	0.05
Therapeutic plasma exchange	15%	26%	0.14

IQR: interquartile range, CRRT: continuous renal replacement therapy, ECMO: extracorporeal membrane oxygenation, ATD: anti-thyroid drugs, BWS: Burch–Wartofsky point scale.

**Table 4 diagnostics-13-03028-t004:** Strength and Quality of evidence of thyroid storm treatment based on the Japanese guidelines.

Measures of Treatment	Strength of Recommendation	Quality of Evidence
Antithyroid drugs (ATDs)	High	Low
Inorganic iodide	High	Moderate
Corticosteroids	High	Moderate
Cooling with acetaminophen and mechanical cooling	High	Low
Therapeutic plasmapheresis	Weak	Low
Central nervous system manifestations treatment	Strong	Low
Tachycardia treatment	High	Low
Atrial fibrillation treatment	High	Low
Acute congestive heart failure	High	Low

**Table 5 diagnostics-13-03028-t005:** Pharmacological treatment options: cases, mechanism of action, indication and adverse events.

Treatment Modalities	N of Cases	Doses	Mechanism of Action/Indications	Side Effects and Contraindications
**Anti-thyroid drugs** (ATD) Carbimazole (CBZ) Methimazole (MMI) Propylthiouracil (PTU)	228	- MMI and CBZ oral 20–30 mg/day every 6–4 h. - PTU: 200 mg every 4 h.	First line of treatment to control TS. Propylthiouracil/Thionamides: T3 production and release blockers. Iodine: Inhibition of preformed thyroid hormones. Propylthiouracil and steroids: Reduce peripheral T3 to T4 conversion, prevent TS if suspected.	**Agranulocytosis.** Neutropenia. **Hepatic dysfunction or failure.** Cholestatic liver injury. Transaminitis. Seen from days to weeks, particularly with PTU. Renal dysfunction or injury. **Multiple organ failure.** **Rash.** **Thrombocytopenia** (CBZ may be switched to PTU). **Antineutrophilic cytoplasmic** antibody vasculitis (PTU). **Antithyroid arthritis** syndrome (CBZ/MMI).
Inorganic iodide Saturated solution of Potassium iodide (SSKI) Lugol iodine	111	SKKI: 200 mg/day. Lugol Iodine: 5–10 drops orally once in 6–8 h.	Wolff–Chaikoff effect Transient reduction in thyroid hormones. Fast acting in comparison to ATDs and CS. - Given 1 h after ATD, to prevent further thyroid hormone release. - Decreases blood flow to thyroid gland and so can be given prior to thyroidectomy.	**Hyperkalemia** (potassium iodide). **Due to the transient action**: In short-term use, decrease dose prior to tapering ATDs dose. Long-term administration (2–12 weeks) may cause hypersecretion of thyroid hormones.
Cholestyramine	33	A total of 4 g oral intake 2–4 times a day.	- Elimination of thyroid hormone in enterohepatic circulation by binding to iodothyronines. - Indications: Prevent refractory-induced hyperthyroidism. Iodine-induced hyperthyroidism. In case of ATD contraindication.	
**Corticosteroids** Hydrocortisone/Dexamethasone prednisone	172	-IV/IM hydrocortisone: 150. mg/day every 6 h. -IV dexamethasone; 2 mg every 6 h.	- When given in high doses, it inhibits thyroid hormone release, T4 and T3 conversion inhibition, and prevents adrenal insufficiency related to the hypermetabolic state of TS. - Increases vasomotor stability. - Given until TS resolves.	
**Beta Blockers** Propranolol (NCBB) Metoprolol Esmolol (SC) Bisoprolol Landiolol (USC) Sotalol	191	-Propranolol: 1. oral or NGT 60–80 mg, 2. IV: 0.5–1 mg over 10 min followed by 1–2 mg over 10 every few hours. -Short-acting (Esmolol): a loading dose of 250–500 mcg/kg, followed by 50–100 mcg/kg infusion.	Blocking peripheral conversion of inactive T4 to active T3. Control of the hyperadrenergic state and peripheral symptoms. Intravenous infusion CBB. Dosage is controlled meaning that cessation can be instantly carried out; this prevents the occurrence of CS. **Indications** Left ventricular dysfunction. Anxiety. Tachycardia. Hypertension. Tachyarrhythmia control. Landiolol for AHF and AF.	**Cardiogenic shock** ATDs should be used instead if the aim is to decrease conversion of T4 to T3. Bisoprolol is recommended over propranolol for tachycardia. **Pulseless electrical activity.** **Circulatory collapse.** **Hypotension.** **Refractory hypotension** - Bronchoconstriction with bisoprolol.
**Calcium channel blockers** Verapamil Diltiazem	30	IV diltiazem push: 20 mg.	- Inhibit Ca^2+^ into excitable cells, resulting in smooth muscle dilation. - Negative inotropes in cardiac cells. - Indications: Ventricular rate control. Atrial fibrillation. Cardioversion. - Calcium channel blockers may be used if BB are contraindicated. - Was given for AF prior to TS diagnosis then discontinued when diagnosis made.	- Cardiogenic shock. - Asystole.
**Digoxin**	25	IV: 0.125–0.25 mg.	Increases cardiac contractility as it binds and inhibits the Na/K-ATPase pump within cardiac myocytes. Positive inotropic effect: Marked tachycardia. Atrial fibrillation increased ventricular response. Heart failure.	Avoid in case of renal dysfunction as it increases renal clearance. - Worsening hypotension.
Inotropes (Vasopressors) Dopamine Dobutamine Epinephrine Levosimendan Noraderanline Milrinone	81	Dobutamine: infusion 2 (ug/kg/min) Noradrenaline.	**Dobutamine/dopamine**: Inotrope with high affinity to B1 adrenergic receptors. Circulatory support in CS. Restore sinus rhythm. **Levosimendan**, calcium sensitizer and phosphodiesterase-3 inhibitor. **Milrinone**: Improves contractility. Ease VA-ECMO weaning.	Refractory cardiogenic shock. Dobutamine’s ineffectiveness is seen when given with high doses of propranolol.
**Amiodarone**	19	IV: 125 mg over 10 min followed by a 0.8 mg infusion for 6 h.	- An iodine-rich class III antiarrhythmic - Blocks 5′mono-deiodination of t4 in peripheral tissues as the liver and pituitary gland. Serum T3 decreases while serum T4 slightly increases. TSH remains unaffected. - Increases action potential duration and prolongs the effective refractory period within myocytes through blocking potassium channels. - Most common antiarrhythmic in ICU due to stable properties. Refractory tachyarrhythmia. Cardioversion: ventricular rate control. Controls TS. Prevention of arrhythmia. Refractory MAT associated TS multi focal atrial tachycardia.	- Hyperthyroid activity and thyrotoxic precipitant (Jod- Basedow phenomenon). - Amiodarone-induced thyrotoxicosis. - Hepatotoxicity; worsened ischemic hepatic failure. - Worsening hypotension

**Table 6 diagnostics-13-03028-t006:** Indications and complications of non-pharmacological treatment of thyroid storm.

Mechanical Modality	Cases	Mechanism of Action/Indications	Side Effects and Contraindications
Extracorporeal membrane oxygenation (ECMO)	40	Severe hypotension. Circulatory collapse (PEA). Cardiogenic shock. Maintains stability in patients until euthyroid state is achieved. Increased blood lactate level. Reversible severe cardiac or pulmonary failure unresponsive to conventional therapy.	Patient condition may be incompatible with life after being off ECMO. Age and size. Preexisting comorbidities. ECMO insertion may cause: -Bleeding. -Thromboembolism. -Strokes. Access injuries: -Hemorrhage. -Arterial dissection -Distal ischemia
Therapeutic plasma exchange (TPE)	40	Rapid decline in thyroid hormones where a new colloid replaces thyroglobulin and bound TH as they are removed from circulation. Hepatic dysfunction where ATD must be discontinued. Prepares patient to be fit enough to undergo thyroidectomy.	Bleeding and infection due to TPE’s removal of IG and clotting factors. Hemodynamic instability. Transfusion reaction. Nausea and vomiting. Respiratory distress. Seizure.
Continuous renal replacement therapy (CRRT)	25	Acute kidney injury (AKI). Hyperkalemia secondary to thyrotoxicosis-induced rhabdomyolysis. Metabolic acidosis. Rapid control over body temperature in case of hyperthermia.	Hypothermia.
Continuous ven o-venous hemodialysis (CVVHD)	10	Used after patients developed ischemic hepatitis, coagulopathy, and anuric kidney injury with metabolic acidosis.	Hypotension leading to uncontrollable tachyarrhythmias. Technical difficulties with initiation of TPE which delayed its initiation.
Biventricular Impella Device	5	Increases cardiac output in cases of biventricular failure. Acute reversible cardiogenic shock causes.	Long-term use causes: Hemolysis. Thrombocytopenia.
Left ventricular Assist Device (LVAD)	3	LVAD is a treatment of heart failure (INTERMACS). It is also the bridge to transplant, the bridge to decision, the bridge to recovery and destination to therapy. Before LVAD implantation right ventricle hemodynamics should be estimated.	LVAD may causes right-side HF as it causes bowing of the interventricular septum which affects RV contraction; thus, RVAD is placed.
Intra-Aortic balloon pump (IABP)	23	To maintain hemodynamic stability. Circulatory support.	May be unsatisfactory and replaced with ECMO.
Thyroidectomy	46	- Due to thyroid dysfunction, patient developed liver dysfunction which had a high risk of transition to fulminant hepatitis. - Refractory thyrotoxicosis. - Renal and liver dysfunction. - Decompensated heart failure. - If patient wishes to be pregnant in the future. - Amiodarone-induced TS. - Refractory cardiogenic shock and repeated cardiac arrest. **Effects of treatment**: - Resolution of severe symptoms. - Increased ejection fraction. - Improved hyperbilirubinemia. - Ellipsoid heart geometry. - Euthyroid with replacement. - Sinus rhythm restored. - Improved coagulation.	
Radioactive iodine ablation/therapy (RAI)	8	Definitive treatment but requires lifelong levothyroxine.	Permanent impact: -Hypocalcemia. -Recurrent laryngeal nerve damage. Hemorrhage necessitating reoperation.
Extracorporeal albumin dialysis (continuous and Single-pass albumin dialysis (SPAD)	1	Liver support therapy and detoxification: -In liver failure. -Hepatorenal syndrome. Thyrotoxicosis. High-capacity hormone removal. -Due to the long period of usage in SPAD, it is superior to plasmapheresis in terms of the number of hormones removed.	

## Data Availability

Not applicable.

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
