# Peer review of "Data Analysis and Systematic Scoping Review on the Pathogenesis and Modalities of Treatment of Thyroid Storm Complicated with Myocardial Involvement and Shock"

_diagnostics, 2023, doi:10.3390/diagnostics13193028_

Round 1

Reviewer 1 Report

Dear Editor, Thank you very much for the opportunity to review this interesting article. Despite the many question marks that arise when reading the manuscript, the article is generally well written and deals with an extremely important topic. I believe that the article will be well received by the readers. Therefore, I believe that the article deserves publication.

Author Response

Thank you for very positive feedback

Reviewer 2 Report

Thank you very much for an opportunity to read and evaluated very interesting review written by Eman Elmenyar et al

I have only one comment:

1.     Table 6 : Indication and complication of non-pharmacological treatment  of thyroid storm- LVAD is a treatment of heart failure (INTERMACS 4,5 ->2), and is also bridge to transplant, bridge to decision , bridge to recovery and destination therapy .

Before LVAD implantation you must estimate right ventricle (hemodynamic, echo, and laboratory test parameters - 30% patients suffer from RHF in 30 days after LVAD placement because of RHF before implantation. LVAD doesn’t cause RHF if you estimate all parameters. RHF is one of the contraindications for LVAD which may lead to inadequate inflow in the device. In some circumstances, the use of RV mechanical support may be required.

Author Response

Thanks you and the LVAD info amended in table 6 as requested

Reviewer 3 Report

The paper titled “Data Analysis and Systematic Scoping Review on The pathogenesis and Modalities of Treatment of Thyroid Storm Complicated with Myocardial Involvement and Shock” highlights the role of thyroid storm and its complication in terms of myocardial affection, tachyarrhythmia, and cardiogenic shock. They did perform a systematic analysis and summarized the data of 72 female and 52 male patients. The content is very extensive, and study is potentially interesting, but there are several issues that need to address.

Major Comments –

1.      The introduction section should be made more concise. Would it be possible to include few more specifics regarding Thyrotoxicosis. 2) Discuss about Hyperthyroidism. 3) From Discussion section remove pathophysiology and Definition of thyroid storm and its Diagnostic Criteria.

2.      In the introduction first paragraph authors discuss about “free hormone”. Can they please say which free hormone are they referring to free fraction/ nonbound fraction (“available free hormone in the circulation and its exaggerated”).

3.      Did authors choose Co-morbidities of patients diagnosed with thyroid storm and comparing with cardiovascular events.

4.      The discussion section should be made more concise.  The prime goal of systematic review is to summarize the major findings from the review and then discuss about limitations of the study and reliability of the results. They have provided extensive information and looks like standard review.

5.      The inclusion and exclusion criteria of study design is not well defined. Here is an example “A case of thyroid storm with cardiac arrest” Yutaka Nakashima,1 Tsuneaki Kenzaka,2 Masanobu Okayama,3 and Eiji Kajii3. A case of thyroid storm with cardiac arrest - PMC (nih.gov). Why is this study not considered?

Minor comments –

1.      The whole manuscript needs language editing in order to make it easier to read and follow. Please, stay consistent with the style.

2.      Please check page 28 outcome and prognosis section – “thi systemaqtic” scoping review

3.      Please check supplementary materials section as Table S1 is missing.

4.      In tabular 6, what is the difference between “ and ○”.

5.      On page 25 check the spelling “shoiwed" and hifghrt.

Major editing is needed. 

Author Response

I would like to thank you

The introduction section should be made more concise. Would it be possible to include few more specifics regarding Thyrotoxicosis. 2) Discuss about Hyperthyroidism. 3) From Discussion section remove pathophysiology and Definition of thyroid storm and its Diagnostic Criteria

Reply: introduction revised as requested thanks.

  1. In the introduction first paragraph authors discuss about “free hormone”. Can they please say which free hormone are they referring to free fraction/ nonbound fraction (“available free hormone in the circulation and its exaggerated”).

Reply: sentence revised, free nonbound fraction of the thyroid hormone thanks.

  1. Did authors choose Co-morbidities of patients diagnosed with thyroid storm and comparing with cardiovascular events.

Reply: Thanks, but could you clarify your request as I could not get it. However, if you mean comorbidities as DM, HTN, most of the cases did not describe such details

  1. The discussion section should be made more concise.  The prime goal of systematic review is to summarize the major findings from the review and then discuss about limitations of the study and reliability of the results. They have provided extensive information and looks like standard review.

Reply: thanks, we revised the discussion, however, this is not a systematic review but it is a systematic form of the scoping review, we added this clarification at the end of the limitation section

  1. The inclusion and exclusion criteria of study design is not well defined. Here is an example “A case of thyroid storm with cardiac arrest”Yutaka Nakashima,1 Tsuneaki Kenzaka,2 Masanobu Okayama,3 and Eiji Kajii3. A case of thyroid storm with cardiac arrest - PMC (nih.gov). Why is this study not considered?

Reply: the eligibility criteria revised and we included more cases therefore we reached 256 cases to be sure we did not miss any cases till august 2023.

Minor comments –

  1. The whole manuscript needs language editing in order to make it easier to read and follow. Please, stay consistent with the style.

Reply: done, language editing done through professional editing services

  1. Please check page 28 outcome and prognosis section – “thi systemaqtic” scoping review

Reply: corrected, thanks

  1. Please check supplementary materials section as Table S1 is missing.

Reply: suppl file added

  1. In tabular 6, what is the difference between “ and ○”.

Reply: corrected, thanks

  1. On page 25 check the spelling “shoiwed" and hifghrt.

Reply: corrected , thanks

Round 2

Reviewer 3 Report

The authors have addressed all the Comments. It is an excellent review and needs to be published. 

Author Response

  • The references are not uniform. Some are with authors, year & others are journal title, year. Please be consistent.

Reply: done

2-  Line 87 - replace insignificant with 'undetectable'; TSH units should be mU/L not Pmol/L. (This error was also made in the article you cited - reference 11). The TSH value should be <0.05 mU/L.

Reply: done, thanks

3- In Table 1 it is odd that all the studies you cited used "BNP". Surely some would have used NTproBNP. If this is unclear from the cited literature it would be better to express it as 'high serum natriuretic peptide level' rather than high BNP serum level.

Reply: revised

4- It is also strange that 112/239 subjects had any acute heart failure yet elevated serum 'BNP' occurred in 56/256 subjects only. Please explain

Reply: thanks for raising this point.  We added in table 1 this clarification (*Heart failure was considered if the author mentioned “heart failure” “pulmonary edema”, “CHF”, “diastolic HF”, “systolic dysfunction” with or without reporting LV ejection fraction percent or serum natriuretic peptide level as the latter two were not done/reported in all the cases .

5- Editing English language was done in the last revision through professional editing services (Paperpal Preflight)